Journal of
**open** psychology data

# The AMATUS Dataset: Arithmetic Performance, Mathematics Anxiety and Attitudes in Primary School Teachers and University Students

DATA PAPER

**KRZYSZTOF CIPORA**

**MARISTELLA LUNARDON**

**NICOLAS MASSON**

**CARRIE GEORGES**

**HANS-CHRISTOPH NUERK**

**CHRISTINA ARTEMENKO**

*Author affiliations can be found in the back matter of this article

]u[ ubiquity press

## ABSTRACT

Although the mathematics anxiety-performance link has been extensively studied, its interplay with other emotional and attitude constructs is still unclear. The present dataset includes measures of mathematics anxiety and arithmetic performance alongside different types of anxiety (i.e., state, test, general anxiety and neuroticism), attitudes towards mathematics (i.e., mathematics self-concept, mathematics self-efficacy and mathematics-gender stereotype endorsement), and math-unrelated constructs among university students ($N = 848$) and primary school teachers ($N = 258$) from Germany and Belgium. The data is accessible in the Open Science Framework (https://osf.io/gszpb/).

**CORRESPONDING AUTHOR:**

**Krzysztof Cipora**

Centre for Mathematical Cognition, Loughborough University, Loughborough, United Kingdom; University of Tuebingen, Germany

k.cipora@lboro.ac.uk

**KEYWORDS:**
arithmetic performance; mathematics anxiety; mathematics self-concept; university students; primary school teachers

**TO CITE THIS ARTICLE:**
Cipora, K., Lunardon, M., Masson, N., Georges, C., Nuerk, H.-C., & Artemenko, C. (2024). The AMATUS Dataset: Arithmetic Performance, Mathematics Anxiety and Attitudes in Primary School Teachers and University Students. *Journal of Open Psychology Data,* 12: 10, pp. 1–17. https://doi.org/10.5334/jopd.115

# (1) BACKGROUND

Arithmetic skills, i.e., the ability to solve operations such as additions, subtractions, multiplications, and divisions, are fundamental for everyday activities, particularly within educational settings where they are actively taught and learned. People vary in their levels of arithmetic proficiency, and, to understand these interindividual differences, researchers have focused on emotions and attitudes linked to mathematics learning.

## 1.1 THE IMPORTANCE OF MATHEMATICS ANXIETY

For many individuals, engagement in mathematics activities is accompanied by feelings of tension and anxiety specific to number manipulation and problem-solving (Cipora et al., 2022). More than a negative feeling, mathematics anxiety has negative consequences, such as mathematics avoidance. Indeed, mathematics anxiety can influence young adults' career paths by dissuading them from pursuing university programs with dense mathematical content (Ahmed, 2018; Beilock & Maloney, 2015; Schmitz et al., 2023). Even students proficient in mathematics, often drawn to pursue Science, Technology, Engineering, and Mathematics (STEM) disciplines, are not immune to mathematics anxiety (Betz, 1978). For instance, higher mathematics anxiety predicts lower grades and reduced enrolment in STEM courses, irrespective of individual mathematics proficiency (Daker et al., 2021).

## 1.2 THE ROLE OF TEACHERS

Mathematics anxiety is not just an adult's phenomenon but can be traced back to primary school, when children enter formal mathematics education (Ramirez et al., 2013; Wu et al., 2012; Krinzinger et al., 2009). At this stage, teachers play a pivotal role, as they serve as role models for their students and can significantly influence their attitudes (e.g., Blazar & Kraft, 2017). Notably, the elevated mathematics anxiety reported among primary school teachers compared to other adult groups raise concerns about negative role model learning (Artemenko et al., 2021; Hembree, 1990; Kelly & Tomhave, 1985; Çatlıoğlu et al., 2014; Uysal & Dede, 2016). Teacher's mathematics anxiety may induce mathematics anxiety in their students (e.g., Richland et al., 2020) as well as impact their mathematics achievement (e.g., Beilock et al., 2010; Ramirez et al., 2018). The negative consequences that teachers' mathematics anxiety has on their pupils' mathematics attitudes and performance requires further investigations.

## 1.3 INDIVIDUAL DIFFERENCES IN MATHEMATICS ANXIETY AND RELATED CONSTRUCTS

Mathematics performance, as well as educational choices, are not solely determined by mathematics anxiety. Other individual factors contribute to this interplay.

### 1.3.1 Other forms of anxiety

Common (trait) mathematics anxiety is associated with other forms of anxiety (Cipora et al., 2015; Hembree, 1988, 1990; Lunardon et al., 2022; Orbach et al., 2020; Rossi et al., 2023). These other forms of anxiety are also associated with mathematics performance, although these associations are weaker compared to that of mathematics anxiety. Importantly, the influences of mathematics anxiety usually prevail, when other forms of anxiety are included in a model (Demedts et al., 2022; Hill et al., 2016). Moreover, mathematics anxiety can coexist with other forms of anxiety in some individuals, while others may experience specific academic anxiety types (i.e., test and mathematics anxiety), which have distinct relations to arithmetic performance (Carey et al., 2017; Mammarella et al., 2018; Rossi et al., 2023). The forms of anxiety considered in this context are:

- *state anxiety*: experienced at the very moment of performing a task (Endler & Kocovski, 2001)
- *test anxiety*: experienced in evaluative settings in general (Hembree, 1988)
- *general anxiety*: the general tendency to feel anxious about everyday situations (Hembree, 1990)
- *neuroticism*[1]: the broader tendency to be emotionally unstable (Cipora et al., 2015; Lunardon et al., 2022; Rossi et al., 2023)

### 1.3.2 Positive attitudes

Individual beliefs and stereotypes regarding personal attitudes and skills, especially in the mathematical domain, can interact with mathematics anxiety. Positive mathematics attitudes negatively correlate with mathematics anxiety (Hembree, 1990; Macmull & Ashkenazi, 2019; Rossi et al., 2022) and positively influence mathematics performance (Marsh et al., 2006; Pajares & Miller, 1994) – in contrast to mathematics anxiety. Relevant attitudes in this context are:

- *academic self-concept*: the degree to which people perceive themselves as proficient in specific academic domains, such as mathematics and language (Goetz et al., 2007)
- *mathematics self-efficacy*: the degree of confidence in one's own capabilities to solve mathematical tasks (Bandura, 2012)
- *subject liking and persistence*: intrinsic enjoyment experienced when completing a task and the continuous effort despite its difficulty (Pintrich & Schunk, 2002)
- *mathematics-gender stereotype endorsement*: the degree of agreement with the false belief that mathematics is for men and not for women (Blanton et al., 2002)

### 1.4 OBJECTIVES AND THE PRESENT DATASET

For all these reasons, the complex interplay between individual differences in anxiety, mathematics-specific attitudes, and mathematics anxiety warrants further investigation. How do these factors interact and influence arithmetic performance? Do these patterns vary among students pursuing careers with varying degrees of mathematics load? Do primary school teachers exhibit different mechanisms compared to students in other study programs?

The present dataset may offer some insights into these relationships as it includes mathematics anxiety and arithmetic performance, different forms of anxiety (neuroticism, general anxiety, test anxiety and state anxiety), and different attitudes towards mathematics (mathematics self-concept, mathematics self-efficacy, persistence in mathematical tasks, and, in case of pre-service and in-service teachers, attitudes toward mathematics teaching and mathematics-gender stereotype endorsement). Additionally, self-concept, liking, persistence, and teaching attitudes were also measured for non-mathematics subjects, to provide discriminant validity. The sample consists of university students and primary school teachers from Germany and Belgium. The aim was to provide a rich dataset to investigate mathematics anxiety and its link to arithmetic performance and other mathematics attitudes among university students and teachers. The dataset also allows comparing teachers and non-teachers, which is especially important given that most studies are conducted with non-teachers, while teachers are probably the most important role models (see above).

## (2) METHODS

### 2.1 STUDY DESIGN

The data were collected via three web-based studies. Sample 1 involved German university students. Sample 2 and 3 involved pre-service and in-service primary school teachers from Germany and Belgium, respectively. For each sample, the study included one assessment session of approximately 15 minutes.

The survey foci were mathematics anxiety and mathematics attitudes. We assessed different types of anxiety, such as state anxiety, mathematics anxiety, test anxiety, general anxiety, and neuroticism. Additionally, we assessed mathematics self-efficacy, mathematics self-concept, mathematics liking and persistence. Self-concept, liking, persistence, and teaching attitudes were measured for non-mathematics subjects, to provide discriminant validity. Arithmetic performance was also assessed. Additional mathematics teaching attitudes were investigated in teachers (Sample 2 and 3), such as ease and enjoyment of teaching mathematics and mathematics-gender stereotype endorsement.

### 2.2 TIME OF DATA COLLECTION

Data for Sample 1 was collected in June 2017, for Sample 2 in January 2018, and for Sample 3 from May 2018 until December 2018.

### 2.3 LOCATION OF DATA COLLECTION

All studies were conducted online. In Sample 1, respondents were students at the University of Tuebingen (Baden-Wuerttemberg in South-West Germany). Pre-service teachers in Sample 2 were students at the Ludwigsburg University of Education (Baden-Wuerttemberg); in-service teachers were mostly based in Baden-Wuerttemberg. Pre-service teachers in Sample 3 were students at the Haute Ecole Galilée (Brussels, Belgium); in-service teachers were mostly based in Brussels. No information about the participants' residence (urban/rural) is available.

### 2.4 SAMPLING, SAMPLE AND DATA COLLECTION

For each sample, we employed convenience sampling and then excluded participants that did not meet the eligibility criteria described below. Table 1 shows the total number of participants involved in the study and the number of participants excluded from the dataset.

Recruitment strategies for Sample 1 included an internal e-mail to students at the University of Tuebingen; for Sample 2 an internal e-mail to students within the Ludwigsburg University of Education, an e-mail to teachers by a school headmaster, and personal contacts; for Sample 3 an e-mail to current and former students by professors of the Haute Ecole Galilée. As compensation, all participants were offered to enter a raffle for vouchers.[2]

Students in Sample 1 were eligible if they were native German speakers. Participants in Sample 2 and 3 were eligible if they were primary school teachers or studying primary school education to become a teacher. Teachers in primary school had to have a sufficient language level in either German or French to take part. The teachers enrolled in the study were working or preparing to work with children who enter schooling in their respective countries at the age of six. Primary school consists of grades 1–4 in Germany and grades 1–6 in Belgium. In all studies, participants were only included if they were at least 18 years old.

The initial sample consisted of $N = 1404$ (1049 for Sample 1, 164 for Sample 2, and 191 for Sample 3). After excluding participants (see Table 1), the final sample consisted of $N = 1106$ (848 for Sample 1, 131 for Sample 2, and 127 for Sample 3). Table 2 reports demographics for these samples.

### 2.5 MATERIALS/SURVEY INSTRUMENTS

A detailed overview of the measures, instructions, scales, items, and translations can be found in the codebook (see *AMATUS_codebook.xlsx*). Table 3 summarizes for each measure the number of participants who provided

|  | SAMPLE 1 | SAMPLE 2 | SAMPLE 3 | TOTAL |
|---|---|---|---|---|
| Initial sample (*N*) | 1049[a] | 164 | 191 | 1404 |
| Exclusion criteria (*n*) |  |  |  |  |
| Not finished | 110 | 22 | 45 | 177 |
| Not eligible | 34 | 8 | 3 | 45 |
| Noisy environment | 36 | 0 | 12 | 48 |
| Dishonest responses | 4 | 0 | 0 | 4 |
| Duration > 30 min | 17 | 3 | 4 | 24 |
| Final sample (*N*) | 848 | 131 | 127 | 1106 |
| Exclusion criteria for the arithmetic test (*n*) |  |  |  |  |
| Skipping items | 113 | 24 | 57 | 194 |
| Further quality checks without exclusion (*n*) |  |  |  |  |
| Device used for survey: |  |  |  |  |
| tablet | *NA* | 4 | 4 | 8 |
| smartphone | *NA* | 22 | 29 | 51 |
| computer/laptop | *NA* | 105 | 94 | 199 |
| Breaks | 29 | 3 | 6 | 38 |

**Table 1** Exclusion criteria and quality check for the overall sample and separately for the three samples.

*Notes. NA* = not applicable because the question was not included in the survey.

|  | WHOLE SAMPLE | SAMPLE 1 | SAMPLE 2 | SAMPLE 3 |
|---|---|---|---|---|
| *N* | 1106 | 848 | 131 | 127 |
| Characteristic |  | University students (Germany) | Pre-/in-service teachers (Germany) | Pre-/in-service teachers (Belgium) |
| Age *M (SD)* | 24 (5.49) | 23.55 (4.17) | 23.96 (5.89) | 27.06 (10.04) |
| Age range |  |  |  |  |
| <20 years | 99 | 87 | 8 | 4 |
| 20–29 years | 901 | 700 | 109 | 92 |
| 30–39 years | 80 | 56 | 10 | 14 |
| 40–49 years | 13 | 1 | 2 | 10 |
| over 50 years | 13 | 4 | 2 | 7 |
| Females/Males | 74%/26% | 70%/30% | 89%/11% | 84%/16% |
| Mathematics load[a] |  |  |  |  |
| low |  | 387 |  |  |
| medium |  | 342 |  |  |
| high |  | 117 |  |  |
| Mathematics focus/ Non- mathematics focus[b] |  |  | 59%/41% | 61%/7%[c] |
| In-service/ Pre-service[d] |  |  | 21%/79% | 32%/68% |

**Table 2** Demographics for the overall sample and separately for the three samples.

*Notes.* [a]Mathematics load content in students' study program. Two participants were not assigned to a mathematics load category because the study program name they entered could not be traced back to any existent degree course. [b] Percentage of teachers specializing in mathematics or in another subject. [c] 40 participants (32%) reported no main focus. [d] Percentage of participants working as teachers ("in-service") or still in education ("pre-service").

Cipora et al. *Journal of Open Psychology Data* DOI: 10.5334/jopd.115

| MEASURE | N | N ITEMS | RESPONSE SCALE | THEORETICAL RANGE | M (SD) | MIN–MAX | CRONBACH'S α | ORDINAL α | SKEWNESS (SE) | KURTOSIS (SE) | SAMPLES |
|---|---|---|---|---|---|---|---|---|---|---|---|
| Mathematics grade | 1106 | 1 | 1–6[a] | 1–6 | 2.59 (1.29) | 1–6 | NA | NA | 0.49 (0.07) | −0.64 (0.15) | 1–3 |
| Mathematics influence study choice | 1106 | 1 | 1–9 | 1–9 | 5.16 (1.99) | 1–9 | NA | NA | −0.04 (0.07) | −0.19 (0.15) | 1–3 |
| Neuroticism (BFI-K, BFI-Fr) | 1106 | 8 | 1–5 | 8–40 | 23.46 (6.05) | 9–40 | 0.84 | 0.86 | 0.16 (0.07) | −0.62 (0.15) | 1–3 |
| General anxiety (GAD-7) | 1106 | 7 | 1–4 | 7–28 | 12.78 (4.14) | 7–28 | 0.84 | 0.87 | 1.13 (0.07) | 1.03 (0.15) | 1–3 |
| Test anxiety (TAI-short) | 1106 | 5 | 1–4 | 5–20 | 11.72 (3.95) | 5–20 | 0.84 | 0.88 | 0.29 (0.07) | −0.81 (0.15) | 1–3 |
| Mathematics anxiety total (AMAS) | 1106 | 9 | 1–5 | 9–45 | 19.07 (6.88) | 9–45 | 0.89 | 0.93 | 0.67 (0.07) | 0.01 (0.15) | 1–3 |
| Learning mathematics anxiety | 1106 | 5 | 1–5 | 5–25 | 7.62 (3.38) | 5–25 | 0.84 | 0.90 | 1.61 (0.07) | 2.80 (0.15) | 1–3 |
| Mathematics evaluation anxiety | 1106 | 4 | 1–5 | 4–25 | 11.45 (4.21) | 4–20 | 0.88 | 0.90 | 0.05 (0.07) | −0.95 (0.15) | 1–3 |
| Mathematics self-concept (SDQ-III) | 1106 | 4 | 1–4 | 4–16 | 11.35 (3.13) | 4–16 | 0.89 | 0.93 | −0.33 (0.07) | −0.78 (0.15) | 1–3 |
| Language self-concept (SDQ-III) | 1106 | 4 | 1–4 | 4–16 | 13.72 (2.37) | 4–16 | 0.82 | 0.88 | −0.95 (0.07) | 0.30 (0.15) | 1–3 |
| Mathematics self-efficacy (PISA) | 1106 | 6 | 1–4 | 6–24 | 20.06 (3.50) | 8–24 | 0.83 | 0.89 | −0.77 (0.07) | −0.09 (0.15) | 1–3 |
| Liking mathematics | 1106 | 1 | 1–5 | 1–5 | 3.23 (1.31) | 1–5 | NA | NA | −0.23 (0.07) | −1.05 (0.15) | 1–3 |
| Liking science | 1106 | 1 | 1–5 | 1–5 | 3.65 (1.21) | 1–5 | NA | NA | −0.57 (0.07) | −0.63 (0.15) | 1–3 |
| Liking humanities | 1106 | 1 | 1–5 | 1–5 | 3.85 (1.09) | 1–5 | NA | NA | −0.73 (0.07) | −0.26 (0.15) | 1–3 |
| Persistence math | 1106 | 1 | 1–5 | 1–5 | 3.47 (1.18) | 1–5 | NA | NA | −0.46 (0.07) | −0.69 (0.15) | 1–3 |
| Persistence science | 1106 | 1 | 1–5 | 1–5 | 3.56 (1.09) | 1–5 | NA | NA | −0.49 (0.07) | −0.47 (0.15) | 1–3 |

(Contd.)

| MEASURE | N | N ITEMS | RESPONSE SCALE | THEORETICAL RANGE | M (SD) | MIN–MAX | CRONBACH'S α | ORDINAL α | SKEWNESS (SE) | KURTOSIS (SE) | SAMPLES |
|---|---|---|---|---|---|---|---|---|---|---|---|
| Persistence humanities | 1106 | 1 | 1–5 | 1–5 | 3.85 (0.99) | 1–5 | NA | NA | -0.66 (0.07) | -0.09 (0.15) | 1–3 |
| Arithmetic performance | 912 | 40 | Free entry | 0–40 | 13.68 (6.72) | 0–40 | 0.92 | 0.98 | 0.64 (0.08) | 0.59 (0.16) | 1–3 |
| State anxiety (STAI-SKD) | 1106 | 5 | 1–4 | 5–20 | 8.79 (3.23) | 5–20 | 0.86 | 0.90 | 0.96 (0.07) | 0.42 (0.15) | 1–3 |
| Preference teaching language | 217 | 1 | 1–5 | 1–5 | 4.06 (1.04) | 1–5 | NA | NA | -0.92 (0.17) | 0.05 (0.33) | 2–3 |
| Preference teaching math | 213 | 1 | 1–5 | 1–5 | 4.19 (0.90) | 1–5 | NA | NA | -1.09 (0.17) | 1.03 (0.33) | 2–3 |
| Preference teaching science | 200 | 1 | 1–5 | 1–5 | 3.97 (1.09) | 1–5 | NA | NA | -0.97 (0.17) | 0.23 (0.34) | 2–3 |
| Ease of teaching language | 217 | 1 | 1–5 | 1–5 | 3.83 (1.03) | 1–5 | NA | NA | -0.71 (0.17) | -0.09 (0.33) | 2–3 |
| Ease of teaching math | 212 | 1 | 1–5 | 1–5 | 3.85 (0.88) | 1–5 | NA | NA | -0.49 (0.17) | -0.01 (0.33) | 2–3 |
| Ease of teaching science | 199 | 1 | 1–5 | 1–5 | 3.67 (0.93) | 1–5 | NA | NA | -0.40 (0.17) | -0.16 (0.34) | 2–3 |
| Mathematics-gender stereotype endorsement (FSMAS-SF) | 258 | 9 | 1–5 | 9–45 | 41.67 (4.55) | 25–45 | 0.80 | 0.92 | -1.63 (0.15) | 2.14 (0.30) | 2–3 |

**Table 3** List of the study measures.

*Notes.* The table reports the number of observations (N), the number of items, the theoretical range of scores, mean (M) and standard deviation (SD), minimum and maximum, Cronbach's and ordinal alphas (α), skewness and its standard error (SE), kurtosis and its standard error, and in which study(s) the measure was assessed. The order of the list resembles the order in which questionnaires and tests were administered to the participants. NA = not applicable due to a single item only. ªBelgian grades were originally expressed in the Belgian grading system, as numbers from 0 to 10, with 10 being the best grade; Belgian grades were then recoded in the German grading system.

data, the number of included items, descriptive statistics, reliability computed on the present sample (Cronbach's α and ordinal α) and in which sample(s) that measure was collected (see *AMATUS_script_analysis.R*). The study was implemented in German and French on the SoSci Survey platform (Leiner, 2014). The items designed for the purpose of the study (i.e., demographic data, mathematics load, teacher-specific questions, subject liking and persistence, preference, and ease of teaching) were originally formulated in German and then translated into French by a German-French bilingual, and were proof-read by a native French speaker. No time limit was set to complete the survey, except for the assessment of arithmetic performance (see below for the test description).

**Demographic data.** Demographic data were collected for age (in years), sex (male or female), native language (German/French or other), last school mathematics grade (expressed in the German grading system, as numbers from 1 to 6, with 1 being the best grade; Belgian grades were originally expressed in the Belgian grading system, as numbers from 0 to 10, with 10 being the best grade; Belgian grades were then recoded into the German grade system).

**Mathematics load.** In Sample 1, participants were asked to indicate the mathematics load content in their study program in one of three categories: low, medium, or high. For each response option, examples were provided (e.g., computer science, physics and mathematics for the "high mathematics load"; e.g., chemistry, psychology, and economics for the "medium mathematics load"; e.g., literature, history, and pedagogy for "low mathematics load"). To double-check participant selection, they additionally had to name their study program (these responses are not reported in the dataset). If participants studied in more than one study program, they were asked to choose the appropriate category for their major study program. In the case of two major study programs, they selected the study program including a higher amount of mathematics. If their study program was not included in any of these categories, they could choose a fourth option ("other") and indicate their study program.

Furthermore, all three samples were asked to indicate how much and in what manner the mathematics load in the study program influenced their study program choice. This was assessed by a single item ("For the choice of my study program the mathematics load played...") on a 9-point Likert scale (1 = "... a role because I wanted to avoid mathematical subjects", 5 = "... no role", 9 = "... a role because I wanted to take mathematical subjects"). Lower values indicate that the mathematics load mattered in the sense that led them to avoid mathematics courses. Medium values indicate that mathematics load did not play any role in the choice of the study program. Higher values indicate that the

study program was chosen because of the willingness to pursue mathematics courses.

**Neuroticism.** Neuroticism was assessed by the 8 items of the German short version of the Big Five Inventory (BFI-K) (Rammstedt & John, 2005) and the French version of the Big Five Inventory (BFI-Fr) (Plaisant et al., 2010). Each item corresponded to a statement and participants had to rate their agreement on a five-point Likert scale (1 = very incorrect, 5 = very correct). Items 2, 5, and 7 were reverse coded (in the dataset, the reversed score is reported for these items). The neuroticism score is the sum of each item response with higher values corresponding to higher neuroticism. The authors of the scale reported satisfactory psychometric properties, with an internal consistency of α = .85, and a 6-week test-retest reliability of $r_{tt}$ = .80 (Rammstedt & John, 2005).

**General anxiety.** General trait anxiety was assessed with the German version of the Generalized Anxiety Disorder Screener (GAD-7) (Löwe et al., 2008). The German items were translated into French for this study. Despite being a clinical instrument, GAD-7 was successfully used in healthy populations, including mathematics anxiety studies (e.g., Rossi et al., 2023). Participants were asked to indicate how often they had experienced the emotional states described in each of the 7 items during the last two weeks (1 = not at all, 2 = several days, 3 = more than half the days, 4 = nearly every day). The general trait anxiety score is the sum of each item response with higher values corresponding to higher anxiety levels. Reported reliability (internal consistency: α = 0.89) was satisfactory (Löwe et al., 2008).

**Test anxiety**. Test anxiety was assessed by the original English short version of the Test Anxiety Inventory (TAI-short) (Taylor & Deane, 2002), translated into German and French. Participants were asked to indicate on a 4-point Likert scale how much they agreed to 5 items describing states during examinations (1 = not at all, 4 = very). The test anxiety score is the sum of each item response with higher values corresponding to higher anxiety levels. The original English version reported satisfactory psychometric properties (internal consistency: α = 0.87) and a balance of items from the worry and emotionality subscales of the original long version of the TAI (Taylor & Deane, 2002).

**Mathematics anxiety.** Mathematics anxiety was assessed by the Abbreviated Math Anxiety Scale (AMAS) (Hopko et al., 2003). The German translation by Dietrich et al. (2015) was slightly modified for the items to whole sentences in order to adapt the German version to the original English one. For the French-speaking Belgian sample, the AMAS items were translated into French from the original English version. Participants were asked to indicate how anxious they would feel in each mathematics-related situation described in the 9 items using a 5-point Likert scale (1 = little anxious, 5 = very anxious). The AMAS has two subscales: learning

mathematics anxiety and mathematics evaluation anxiety. Total scores, as the sum of single item responses, were calculated for the whole scale and each subscale with higher values corresponding to higher anxiety levels. The original English version had an internal consistency of α = .90 and a 2-week test-retest reliability of $r_{tt}$ = .85 (Hopko et al., 2003). For the original German version, the internal consistency was α = .92 (Dietrich et al., 2015). The AMAS is a widely used instrument for mathematics anxiety assessment (Cipora et al., 2019), has already been used in web-based research (Cipora et al., 2017; Huber & Artemenko, 2021), and has been translated into many languages (for example, Polish: Cipora et al., 2015; Italian: Primi, 2014; Persian: Vahedi & Farrokhi, 2011).

**Mathematics and language self-concept.** Mathematics and language self-concepts were assessed by the mathematics and verbal ability subscales of the German adaptation (Schwanzer et al., 2005) of the Self-Description Questionnaire III (SDQ-III) (Marsh, 1992), respectively. The German items were translated into French for this study. Each scale consisted of 4 statements regarding ability in mathematics and language. Participants were asked to indicate on a 4-point Likert scale how much they agreed with each statement (1 = do not agree at all, 4 = absolutely agree). Items 2 and 4 for mathematics and 1 and 2 for language self-concepts were reverse coded (in the dataset, the reversed score is reported for these items). The mathematics and language self-concept scores were calculated as the sums of the respective item responses with higher values corresponding to higher self-concept. Reported test-retest reliabilities of the German scales were .90 for mathematical ability and .68 for verbal ability in a 6–8 week time interval (Schwanzer et al., 2005).

**Mathematics self-efficacy.** Mathematics self-efficacy was assessed by German and French items from the PISA 2012 study (for German items, see OECD/BIFIE, 2012; for French items, see OECD, 2014). For six items, participants were asked to indicate on a 4-point Likert scale how confident they feel in solving mathematical tasks (1 = not at all confident, 4 = very confident). The mathematics self-efficacy score is the sum of the item responses with higher values corresponding to higher self-efficacy.

**Subject liking and persistence.** Mathematics, science, and humanities liking was assessed by a single item each. Participants were asked to indicate on a 5-point Likert scale how much they agree with the statements "I like math/science/humanities" (1 = do not agree at all, 5 = absolutely agree). Higher scores correspond to higher levels of liking. Persistence in mathematics, science, and humanities was assessed by a single item each (see Cipora et al., 2015, for a similar measure). Participants were asked to mark how fast they get discouraged when solving subject-related tasks on a 5-point Likert scale (1 = I get discouraged very fast, 5 = I am very persistent). Higher scores correspond to higher persistence.

**Arithmetic performance.** Arithmetic performance was assessed by an arithmetic speed test. Participants had to solve as many as possible of 40 arithmetic problems within two minutes (Rossi et al., 2022; 2023). The test was designed similar to the Math4Speed (Loenneker et al., in press) but mixed all basic arithmetic operations (addition, subtraction, multiplication, and division) of varying difficulty (1- to 3-digit-numbers, with/without carrying/borrowing). The arithmetic problems were presented in a fixed random order (constant for all participants). Participants were instructed to solve the problems in the presented order, mentally and without using a calculator. The score for arithmetic performance was operationalized as the number of correctly solved problems. The final score was retained only for participants who did not skip any item (see Quality control).

**State anxiety.** General state anxiety was assessed by the German five-item short scale STAI-SKD (Englert et al., 2011). The German items were translated into French for this study. Participants were asked to describe the current intensity of their emotions on a 4-point Likert scale (1 = not at all, 2 = somewhat, 3 = moderately, 4 = very much). Importantly, the state anxiety questionnaire was administered after the arithmetic performance test. Satisfactory reliability (internal consistency: α = 0.76) was reported and the scale comprises the two anxiety components of worry and emotionality (Englert et al., 2011).

**Teacher-specific questions.** In Samples 2 and 3, participants were asked whether they were teachers or attending a teacher training (yes or no), their current activity (study, internship, primary school teacher, other), for how long they have been teaching in primary schools, and the number of teaching main foci. According to their current activity, teachers were classified as "in-service" or "pre-service".

Moreover, teachers were asked about their specialization by indicating the number and subjects of their main foci among the subjects: mathematics, German, French, English, science (Belgian teachers: science; German teachers: "Sachunterricht" including natural and social science topics – the specific focus needed to be specified), religious pedagogy, art, music, sport, or other (to be specified). For the sake of anonymization, participants' choices were recategorized into one of the following categories: mathematics, literacy, science, and humanities (see the file *AMATUS_ preprocessing.pdf*).

**Preference and ease in teaching.** Preference and ease of teaching mathematics, science and language were assessed by a single item each (e.g., "I like teaching mathematics." for liking and "Teaching mathematics is easy for me." for ease). Participants were asked to

indicate their agreement to these sentences on a 5-point Likert scale (1 = not at all, 5 = absolutely; additional option: not yet taught).

**Mathematics-gender stereotype endorsement**. The endorsement of the mathematics-gender stereotype was assessed by the Fennema-Sherman Mathematics Attitudes Scale – Short Form (FSMAS-SF) by the "Mathematics as a male domain" scale (Mulhern & Rae, 1998). The original English items were translated into German and French for the purpose of the study. This scale is composed of 9 items, consisting of a statement claiming either that mathematics is a male domain or that women and men are equally competent in mathematics. Participants were asked to rate their agreement on a 5-point Likert scale (1 = strongly disagree, 5 = strongly agree). Items 1 to 4 were reverse coded (in the dataset, the reversed score is reported for these items). The score is the sum of each item response with higher values corresponding to higher levels of mathematics-gender stereotype endorsement. The internal consistency of the original version of the "Mathematics as a male domain scale" was α = .85 (Mulhern & Rae, 1998).

## 2.6 QUALITY CONTROL

In line with the recommendations for web-based experiments (Reips, 2002), some quality items were stated at the end of the survey to ensure data quality. Participants were asked whether they had breaks during the completion of the survey and whether they responded honestly. Moreover, participants were asked to rate the environmental noise during the completion of the survey on a 6-point Likert scale (1 = silent, 2 = very quiet, 3 = fairly quiet, 4 = fairly noisy, 5 = very noisy, 6 = extremely noisy). Additionally, in Sample 2 and 3 participants were asked which device they used to complete the survey (Table 1). Finally, the software saved the survey start and end times for each participant, which allowed us to compute the total completion time in minutes.

No responses were missing for the participants that completed the survey.

We excluded from the dataset participants who declared not to have responded honestly, who reported a very noisy or extremely noisy environment, and who took more than 30 minutes (i.e., more than 2 × typical study duration) to complete the survey (listwise exclusion; Table 1). In addition, response patterns in the arithmetic test were checked. As the participants were explicitly instructed to solve the arithmetic problems in order, arithmetic performance of participants who skipped items was excluded (casewise exclusion; Table 1).

Moreover, we computed reliability for each measure which consisted of more than one item, in terms of Cronbach's and ordinal α (Table 3), and correlations on the whole sample (Table 4). Cronbach's and ordinal α were always above .80, indicating a good internal consistency. In addition, validity-related correlations go in the expected direction. For instance, arithmetic performance correlates positively with measures of mathematics attitudes, such as mathematics influence on study program choice, mathematics liking and persistence, mathematics self-concept, mathematics self-efficacy, preference and ease of teaching mathematics. On the contrary, arithmetic performance correlates negatively with mathematics grade (lower values indicate better grade), liking and persistence in humanities and all the anxiety measures. As concerns divergent validity, the correlations with language-related measures (i.e., language self-concept, preference and ease of teaching languages) were not significant.

## 2.7 DATA ANONYMISATION AND ETHICAL ISSUES

Ethical approval was obtained from the Ethics Committee for Psychological Research of the University of Tuebingen (approval number: Cipora_2017_1204_96). All participants gave informed consent via mouse click before starting the survey.

Data collection and analysis were anonymous. Participants were assigned a random numerical ID when analysing the dataset.

The risk of identification of study participants was reduced by removing free-entry responses and categorization in the data preprocessing step: in Sample 1, free-entry responses about the specific study program were used to check the accuracy of the mathematics load selection and then deleted. In Sample 2 and 3, participants that selected "other" in response to the teacher stage question were reassigned to one of the other categories (study, internship, primary school teacher) based on their free-entry responses; the free-entry responses were then deleted. Moreover, teachers' main foci were categorized into mathematics, literacy, science, and humanities. Finally, given the high variability in age in the teacher samples, we added an age range variable in the dataset and removed teachers' age in years.

## 2.8 EXISTING USE OF DATA

To date, the dataset has only been used for a published paper by Artemenko et al. (2021). This paper compared the level of mathematics anxiety of primary school teachers (Sample 2 and 3) to university students from other study programs (Sample 1), and investigated the association of mathematics anxiety with gender, main focus mathematics, experience duration, preference and ease of teaching mathematics.

In addition, there is a paper in preparation, which was preregistered for secondary data analysis (https://osf.io/z5egb/). This study will employ Latent Profile Analysis (LPA) to investigate whether different combinations of anxiety types and math attitudes can differentiate university students in programs with different mathematics load.

Cipora et al. *Journal of Open Psychology Data* DOI: 10.5334/jopd.115

| | 1. | 2. | 3. | 4. | 5. | 6. | 7. | 8. | 9. | 1. | 11. | 12. | 13. | 14. | 15. | 16. | 17. | 18. | 19. | 20. | 21. | 22. | 23. | 24. | 25. |
|---|---|---|---|---|---|---|---|---|---|---|---|---|---|---|---|---|---|---|---|---|---|---|---|---|---|
| 1. Mathematics grade | – | | | | | | | | | | | | | | | | | | | | | | | | |
| 2. Mathematics influence study choice | -.46*** | – | | | | | | | | | | | | | | | | | | | | | | | |
| 3. Liking math | -.57*** | .66*** | – | | | | | | | | | | | | | | | | | | | | | | |
| 4. Liking science | -.38*** | .35*** | .44*** | – | | | | | | | | | | | | | | | | | | | | | |
| 5. Liking humanities | .22*** | -.28*** | -.22*** | -.26*** | – | | | | | | | | | | | | | | | | | | | | |
| 6. Persistence math | -.49*** | .50*** | .70*** | .36*** | -.21*** | – | | | | | | | | | | | | | | | | | | | |
| 7. Persistence science | -.32*** | .27*** | .33*** | .72*** | -.25*** | .53*** | – | | | | | | | | | | | | | | | | | | |
| 8. Persistence humanities | .20*** | -.29*** | -.26*** | -.24*** | .69*** | -.09** | -.06* | – | | | | | | | | | | | | | | | | | |
| 9. Mathematics anxiety total | .48*** | -.43*** | -.59*** | -.36*** | .17*** | -.57*** | -.36*** | .13*** | – | | | | | | | | | | | | | | | | |
| 1. Mathematics anxiety learning | .43*** | -.36*** | -.46*** | -.32*** | .16*** | -.45*** | -.32*** | .12*** | .88*** | – | | | | | | | | | | | | | | | |
| 11. Mathematics evaluation anxiety | .44*** | -.43*** | -.59*** | -.33*** | .16*** | -.57*** | -.34*** | .12*** | .93*** | .64*** | – | | | | | | | | | | | | | | |
| 12. General anxiety | .17*** | -.12*** | -.15*** | -.12*** | .07* | -.19*** | -.14*** | .02 | .35*** | .32*** | .32*** | – | | | | | | | | | | | | | |
| 13. State anxiety | .24*** | -.15*** | -.20*** | -.16*** | .08** | -.23*** | -.17*** | .01 | .43*** | .37*** | .40*** | .45*** | – | | | | | | | | | | | | |
| 14. Test anxiety | .16*** | -.11*** | -.17*** | -.12*** | -.03 | -.18*** | -.12*** | -.04 | .44*** | .30*** | .48*** | .33*** | .37*** | – | | | | | | | | | | | |
| 15. Mathematics self-concept | -.71*** | .62*** | .80*** | .42*** | -.25*** | .70*** | .40*** | -.25*** | -.64*** | -.55*** | -.61*** | -.20*** | -.25*** | -.22*** | – | | | | | | | | | | |

(Contd.)

| | 1. | 2. | 3. | 4. | 5. | 6. | 7. | 8. | 9. | 10. | 11. | 12. | 13. | 14. | 15. | 16. | 17. | 18. | 19. | 20. | 21. | 22. | 23. | 24. | 25. |
|---|---|---|---|---|---|---|---|---|---|---|---|---|---|---|---|---|---|---|---|---|---|---|---|---|---|
| 16. Language self-concept | .14 *** | -.27 *** | -.25 *** | -.19 *** | .37 *** | -.13 *** | -.07 * | .40 *** | .06 | .03 | .07 * | -.07 * | -.08 ** | -.15 *** | -.18 *** | – | | | | | | | | | |
| 17. Mathmstics self-efficacy | -.39 *** | .38 *** | .50 *** | .43 *** | -.22 *** | .48 *** | .41 *** | -.15 *** | -.52 *** | -.45 *** | -.49 *** | -.21 *** | -.30 *** | -.23 *** | .55 *** | -.10 *** | – | | | | | | | | |
| 18. Neuroticism | .09 ** | -.13 *** | -.17 *** | -.11 *** | .05 | -.24 *** | -.18 *** | -.02 | .34 *** | .24 *** | .37 *** | .60 *** | .34 *** | .37 *** | -.19 *** | -.10 *** | -.26 *** | – | | | | | | | |
| 19. Preference teaching language | .27 *** | -.30 *** | -.34 *** | -.26 *** | .29 *** | -.26 *** | -.15 * | .31 *** | .29 *** | .21 ** | .30 *** | .07 | .08 | .11 | -.35 *** | .47 *** | -.30 *** | .16 * | – | | | | | | |
| 20. Preference teaching math | -.43 *** | .54 *** | .61 *** | .23 *** | -.08 | .44 *** | .17 * | -.12 | -.46 *** | -.37 *** | -.45 *** | -.16 * | -.28 *** | -.27 *** | .57 *** | -.05 | .38 *** | -.18 ** | -.14 * | – | | | | | |
| 21. Preference teaching science | -.23 *** | -.03 | -.06 | .51 *** | .06 | -.05 | .44 *** | .18 * | -.03 | -.05 | -.02 | .06 | -.05 | -.08 | .01 | -.01 | .08 | -.02 | .04 | .13 | – | | | | |
| 22. Ease of teaching language | .26 *** | -.27 *** | -.25 *** | -.24 *** | .24 *** | -.18 ** | -.16 * | .25 *** | .21 ** | .15 * | .22 ** | -.02 | .04 | .07 | -.28 *** | .47 *** | -.20 ** | .02 | .75 *** | -.18 ** | -.12 | – | | | |
| 23. Ease of teaching math | -.35 *** | .37 *** | .57 *** | .24 *** | -.11 | .48 *** | .21 ** | -.11 | -.46 *** | -.37 *** | -.45 *** | -.15 * | -.24 *** | -.22 ** | .54 *** | .01 | .41 *** | -.19 ** | -.19 ** | .71 *** | .02 | -.10 | – | | |
| 24. Ease of teaching science | -.17 * | -.03 | .02 | .52 *** | .04 | .02 | .45 *** | .15 * | -.12 | -.13 | -.09 | -.01 | -.09 | -.13 | .06 | .04 | .14 * | -.14 * | .02 | .12 | .74 *** | -.02 | .25 *** | – | |
| 25. Mathematics-gender stereotype endorsement | -.11 | .14 * | .20 ** | .06 | .14 * | .19 ** | .09 | .19 ** | -.18 ** | -.18 ** | -.14 * | -.02 | -.10 | -.11 | .18 ** | .22 *** | .16 ** | .01 | .10 | .27 *** | .05 | .09 | .25 *** | .04 | – |
| 26. Arithmetic performance | -.22 *** | .24 *** | .32 *** | .16 *** | -.12 *** | .26 *** | .14 *** | -.10 ** | -.28 *** | -.22 *** | -.29 *** | -.09 ** | -.18 *** | -.09 ** | .32 *** | .003 | .36 *** | -.13 *** | -.11 | .33 *** | -.09 | -.004 | .36 *** | -.04 | .25 *** |

**Table 4** Correlations between the study variables. Significant correlations are marked with asterisks directly under the correlations. Significance levels are specified as follows: * $p < .05$; ** $p < .01$; *** $p < .001$. Note that some correlations are very small.

## (3) DATASET DESCRIPTION AND ACCESS

### 3.1 REPOSITORY LOCATION
https://osf.io/gszpb/

### 3.2 OBJECT/FILE NAME
The repository contains the following files:

- *AMATUS_dataset.csv* – The dataset was preprocessed according to the inclusion and exclusion criteria.
- *AMATUS_codebook.xlsx* – The codebook describes all variables and values of the dataset.
- *AMATUS_preprocessing.pdf* – Description of the preprocessing steps from the three raw datasets to the final published dataset.
- *AMATUS_script_preprocessing* – This subfolder contains R scripts for the preprocessing of raw data (*Preprocessing.R* and *AMATUS_script_preprocessing.R*) that resulted in the current dataset. These scripts are shared for inspection but cannot be run on the current dataset.
- *AMATUS_script_analysis.R* – R scripts for analysis run on the preprocessed data (demographics, descriptives, reliability, and correlations).
- *AMATUS_Table4_correlations.csv* – The tables reported in this article, in .csv format.

### 3.3 DATA TYPE
Partially processed primary data. Raw data were processed for the purposes of anonymization, translation, and exclusion of inadequate data (see sections 2.4, 2.6 and 2.7).

### 3.4 FORMAT NAMES AND VERSIONS
The dataset and the correlation table are accessible in non-proprietary .csv format, which can be read by the majority of spreadsheet and text editors. The codebook is in .xlsx format, which can be read with Microsoft Excel, Google Sheets or LibreOffice Calc. The file *AMATUS_preprocessing.pdf* can be opened with any .pdf reader. The files *AMATUS_script_preprocessing.R*, *Preprocessing.R* and *AMATUS_script_analysis.R* can be run with the R software and opened with any text editor.

### 3.5 LANGUAGE
Data and documentation are stored in English. The German and French versions of the items are also available in the codebook.

### 3.6 LICENSE
Data and documentation are licensed under a CC-BY-4.0 license.

### 3.7 LIMITS TO SHARING
The AMATUS data are not under embargo and are fully shared.

### 3.8 PUBLICATION DATE
Data and documentation were published on the 07/06/2024.

### 3.9 FAIR DATA/CODEBOOK
The AMATUS dataset is stored on OSF and is openly accessible.

A codebook is included, where all variables are listed. The codebook indicates the constructs assessed, the instruments used, the variable names in the dataset, a description in English of the variables, the type of variables, whether the item score was reversed, the English version of the items, the used German and French versions of the items, and in which sample each variable was assessed.

To facilitate reusability, when required, item scores were reverse-coded, so that item scores are consistent in the present dataset. Additionally, total scores are reported. For compatibility, the data is saved in .csv format.

## (4) REUSE POTENTIAL

The AMATUS dataset offers a valuable resource for researchers in psychology and mathematics education, enabling the exploration of hypotheses concerning the emotional and motivational aspects of mathematics in adults. Its comprehensive collection of variables addressing mathematics attitudes and different forms of anxiety facilitates an in-depth examination of individual differences in mathematics anxiety and its relation to arithmetic performance.

Researchers could consider examining differences in the mathematics anxiety-performance link depending on different groups of individuals, e.g., university students from study programs with different mathematics load. These groups likely differ in mathematics anxiety, mathematics attitudes, mathematics performance, and their interrelation (Ahmed, 2018; Beilock & Maloney, 2015; Schmitz et al., 2023). In research on personality psychology, different study programs may also correlate with trait-level differences (Vedel, 2016). Thus, the interplay between mathematics-specific and trait variables, and how the educational context influences these relationships, is yet to be fully explored.

Moreover, this dataset could be used to analyze the relationship between teachers' beliefs and the endorsement of mathematics-gender stereotypes in comparison to other university students. Given that some teachers were found to experience higher mathematics anxiety than other groups (Artemenko et al., 2021, Hembree, 1990; Kelly & Tomhave, 1985), it is plausible that they may also exhibit lower mathematics self-concept and self-esteem, which may have repercussions on their pupils' personal beliefs. Exploring teachers' endorsement of mathematics-gender stereotypes could shed light on this phenomenon, particularly given that

Cipora et al. *Journal of Open Psychology Data* DOI: 10.5334/jopd.115

female students typically report higher mathematics anxiety than their male peers (e.g., Hill et al., 2016).

However, what has not been done at all so far, are multivariate analyses like factor, cluster, or discriminant analyses, to identify the underlying structure of these somewhat related constructs. Additionally, for more complex multivariate relations between constructs, path analyses or structural equation modelling could be used to shed better light on the relations of latent constructs to mathematics anxiety. The strength of this dataset lies in its inclusion of a wide array of constructs that can elucidate the mechanisms through which mathematics anxiety influences mathematics behaviour. In addition to well-studied constructs such as mathematics self-concept, mathematics self-efficacy and different types of anxieties, it also incorporates less investigated factors like neuroticism, as well as measures related to university study program choice, persistence, preference and ease of mathematics teaching. This dataset can therefore also be used for future study planning for an informed power analysis: researchers can use this dataset to estimate correlations with relatively narrow confidence intervals or path coefficients in structural models, given the large sample size. Furthermore, the dataset includes measures allowing for convergent and also discriminant validity assessment: persistence, liking, preference and ease of teaching were inquired not only for mathematics, but also for science and humanities and we assessed self-concept both in the mathematics and in the language domain. This enables researchers to study the specificity of constructs such as mathematics anxiety in the context of related constructs.

Some limitations of the AMATUS dataset should also be acknowledged. Firstly, besides the relatively large variance due to the large sample size, self-selection of the sample may have occurred, as the focus of the study was mathematics. This might be particularly applicable for students from low mathematics load study programs and teachers not specialized in mathematics. Given their higher mathematics anxiety and worse mathematics attitudes, there may be an underestimation of effects, especially in the comparisons between different mathematics load or specializations. Secondly, the data was collected online to reach students from different backgrounds. Online data collection is limited because successful compliance with instructions cannot be assured (e.g., mental arithmetic without any aids). At the same time, participants were not incentivised to perform well. However, comparisons of web-based with lab and field studies are still pending (Cipora et al., 2017) and could further prove the validity of the AMATUS dataset. Note, however, that the data was collected before the COVID-19 pandemic, when online research was not as widespread as it is now in 2024. Thirdly, most of the items administered to Sample 3 were translated from German to French without backward translation.

Although translation was carried out by native speakers, the two versions may not be perfectly equivalent (please note that readers can evaluate it themselves as all translations of our materials are shared together with the data). Finally, the data was collected mainly in Germany and partially in Belgium so that it might be limited to these educational systems, cultural backgrounds, and languages. To generalize the findings based on the AMATUS dataset, we invite researchers to translate and use the measures of this study all over the world to compare the resulting findings and check replicability.

In conclusion, the AMATUS dataset can be used to address various research questions with significant implications for the field of mathematics education. Therefore, researchers can conduct secondary analyses, validation studies, and collaborative works by extending the dataset. Moreover, it is a simple dataset to be used for training purposes in psychology and teacher education.

## NOTES

1   Here we group neuroticism together with other anxiety types to distinguish trait measures from attitudes. However, we acknowledge that neuroticism is a broader personality trait encompassing facets other than anxiety (e.g., Ormel et al., 2013).

2   The email addresses for the raffle were stored separately from the collected data and therefore could not be linked. Entering the raffle was not mandatory, and participants who wanted to remain fully anonymous could simply not provide a contact email address.

## ACKNOWLEDGEMENTS

CA, KC, and HCN are members of the LEAD Graduate School & Research Network [GSC1028], and HCN is furthermore a member of the German Center of Mental Health, Section Tuebingen. NM was supported by a fellowship of Teach@Tuebingen, both of which are funded within the framework of the Excellence Initiative of the German federal and state governments. NM was further funded by grants from the National Research Fund of Luxembourg (FNR, Luxembourg) [FNR-INTER/FNRS/17/1178524] and from the Fonds National de la Recherche Scientifique (FRS-FNRS, Belgium) [PDR-T.0047.18 and 1.B303.21]. CA was further supported by the Tuebingen Postdoc Academy for Research on Education (PACE), by the Ministry of Science, Research and the Arts Baden-Wuerttemberg and the European Social Fund, and by the German Research Foundation (DFG; grant number: 468460838, AR 1500/1-1; grant number: 513458453, AR 1500/2-1). KC is supported by the UKRI Economic and Social Research Council (grant number ES/W002914/1).

We acknowledge support from the Open Access Publication Fund of the University of Tuebingen.

## COMPETING INTERESTS

The authors have no competing interests to declare.

## AUTHOR CONTRIBUTIONS

**Krzysztof Cipora** (*Post-Doc at the University of Tuebingen, Germany*): conceptualization, methodology, resources, project administration, writing – review and editing.

**Maristella Lunardon** (*PhD candidate at the International School for Advanced Studies (SISSA), Italy*): data curation, formal analysis, visualization, writing – original draft, writing – review and editing.

**Nicolas Masson** (*Post-Doc at the Université catholique de Louvain, Belgium*): conceptualization, methodology, resources, project administration, supervision, writing – review and editing.

**Carrie Georges** (*Post-Doc at the University of Luxembourg, Luxembourg*): conceptualization, methodology, resources, project administration, supervision, writing – review and editing.

**Hans-Christoph Nuerk** (*Professor at the University of Tuebingen, Germany*): conceptualization, writing – review and editing.

**Christina Artemenko** (*Post-Doc at the University of Tuebingen, Germany*): conceptualization, methodology, resources, project administration, supervision, writing – review and editing.

**Laura Gottwald** (*Student at the University of Tuebingen, Germany*): investigation, project administration.

**Ida Johanna M. L. C. von Lehsten** (*Student at the University of Tuebingen, Germany*): investigation, project administration.

**Stefan Smaczny** (*Student at the University of Tuebingen, Germany*): investigation, project administration.

**Julius Gervelmeyer** (*Student at the University of Tuebingen, Germany*): data curation, formal analysis.

**Andreas Schmitt** (*Student at the University of Tuebingen, Germany*): data curation, formal analysis.

**Tobias Feuerecker** (*Student at the University of Tuebingen, Germany*): data curation, formal analysis.

**Anita Kaiser** (*Professor at the Haute Ecole Galilée of Brussels, Belgium*): investigation.

**Joachim Engel** (*Professor at the Ludwigsburg University of Education, Germany*): investigation.

## AUTHOR AFFILIATIONS

**Krzysztof Cipora** orcid.org/0000-0003-0077-9336
Centre for Mathematical Cognition, Loughborough University, Loughborough, United Kingdom; University of Tuebingen, Germany

**Maristella Lunardon** orcid.org/0000-0002-2316-023X
International School for Advanced Studies (SISSA), IT

**Nicolas Masson** orcid.org/0000-0002-6989-3375
Université Catholique de Louvain, Louvain-la-Neuve, Belgium

**Carrie Georges** orcid.org/0000-0001-7492-7480
Department of Behavioural and Cognitive Sciences, Faculty of Humanities, Education and Social Sciences, University of Luxembourg, Esch-Belval, Luxembourg

**Hans-Christoph Nuerk** orcid.org/0000-0002-0331-7498
Department of Psychology, University of Tuebingen, Tuebingen, Germany; LEAD Graduate School and Research Network, University of Tuebingen, Tuebingen, Germany; German Center of Mental Health, Section Tuebingen, Tuebingen, Germany

**Christina Artemenko** orcid.org/0000-0001-5947-7960
Department of Psychology, University of Tuebingen, Tuebingen, Germany; LEAD Graduate School and Research Network, University of Tuebingen, Tuebingen, Germany

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

## PEER REVIEW COMMENTS

*Journal of Open Psychology Data* has blind peer review, which is unblinded upon article acceptance. The editorial history of this article can be downloaded here:

• **PR File 1.** Peer Review History. https://doi.org/10.5334/jopd.115.pr1

**TO CITE THIS ARTICLE:**
Cipora, K., Lunardon, M., Masson, N., Georges, C., Nuerk, H.-C., & Artemenko, C. (2024). The AMATUS Dataset: Arithmetic Performance, Mathematics Anxiety and Attitudes in Primary School Teachers and University Students. *Journal of Open Psychology Data,* 12: 10, pp. 1–17. https://doi.org/10.5334/jopd.115

**Submitted:** 10 June 2024     **Accepted:** 06 September 2024     **Published:** 17 September 2024

