## [Peer Review History. · Journal of Open Psychology Data]

Dear Dr. Evans,

Thank you for evaluating our manuscript and for the possibility to resubmit it after revision. In the following, you can find our replies to the reviewers' comments and all marked changes in the revised manuscript.

We truly look forward to seeing it published.

Kind regards,
the authors

Reviewer A

Overall impression

Thank you for your willingness to make this dataset publicly available. The variables measured and the size of the dataset provide researchers (and students) an opportunity for a wide variety of analyses that can contribute to a better understanding of this important topic. The gathering of the data, how each variable was measured, how it should be interpreted and the possible uses of the data are well described. There are a few issues that need to be improved.

Reply. Thank you for positive evaluation of our manuscript and the shared dataset. We also wish to thank for your constructive feedback. We reply to all comments below.

Title

1. The title doesn't do justice to the dataset. Consider something like "The AMATUS dataset: Factors affecting the arithmetic performance of students and teachers". Do not use the abbreviation "math" in the title. Related constructs are meaningless in a title. Different mathematical background is not the focus of the dataset.

Reply. We adapted the title as suggested while maintaining the acronym AMATUS (= Arithmetic, Math Anxiety/Attitudes, Teachers, University Students) for our dataset. The new title reads: "The AMATUS dataset: Arithmetic performance, Mathematics anxiety and Attitudes in primary school Teachers and University Students".

Keywords:

2. Consider keywords that truly reflect the value of the dataset and make the article (and therefore, the dataset) discoverable. E.g., arithmetic performance, mathematical anxiety, mathematical attitude, student data, teacher data, etc.

Reply. Thank you for your suggestion. Unfortunately, the journal only allows for 5 keywords. We have chosen the following keywords: "arithmetic performance, mathematics anxiety, mathematics self-concept, university students, primary school teachers"

The word "math":

3. Do not use the word “math” in the article. Always use “mathematics” or “mathematical”, whichever one is appropriate.

Reply. We have replaced the word “math” with “mathematics” or “mathematical” throughout the text.

4. Since “mathematical anxiety” appears often, consider giving it the abbreviation “MA” and refer to MA after first declaring it as the abbreviation of “mathematical anxiety”.

Reply. Thank you for the suggestion. We decided against using the abbreviation “MA” for mathematics anxiety, because this is our main variable and readability of the manuscript decreases.

Grammar:

5. Improve the grammar of the following sentences: “Additionally, endorsing the stereotype that mathematics is for men and not for women correlates with math anxiety and, consequently, with arithmetic performance in a gender-specific manner”. Pre-service teachers in Sample 3 were students at the at the Haute Ecole Galilée (Brussels, Belgium); in-service teachers were mostly based in Brussels in Belgium.

Reply. The first sentence was deleted. We have corrected the second sentence deleting the repetition of “at the” (page 4).

6. Change “teacher specific” to “teacher-specific”

Reply. We have replaced “teacher specific” with “teacher-specific”.

7. Change “easiness of teaching” to “ease of teaching”: several instances in the document

Reply. We have replaced “easiness of teaching” with “ease of teaching” throughout the text and in the codebook.

8. Improve the grammar of “The items designed for the purpose of the study (i.e., demographic data, math load, teacher specific questions, subject liking and persistence, preference and easiness of teaching) were originally formulated in German and then translated into French by a German-French bilingual individual, with the double check of a French native speaker.”

Reply. We have modified the sentence as follows (page 6): “The items designed for the purpose of the study (i.e., demographic data, mathematics load, teacher-specific questions, subject liking and persistence, preference, and ease of teaching) were originally formulated in German and then translated into French by a German-French bilingual individual and were proof-read by a French native speaker.”

9. Change “in” to “on”: The study was implemented in German and French in the SoSci Survey platform (Leiner, 2014).

Reply. We have made the replacement (page 6).

10. Change verbs to “has already been used” and “has been translated”: “The AMAS is a widely used instrument for math anxiety assessment (Cipora et al., 2019), **was already used** in web-based research (Cipora et al., 2017; Huber & Artemenko, 2021), and **was translated** into many languages...”

Reply. We have changed the sentences accordingly (page 13).

11. Improve grammar: “**Liking in** math, science and humanities ...

Reply. We have changed the expression as follows (page 14): “Mathematics, science and humanities liking”.

12. Change to “discriminant”: **discriminance** analyses

Reply. We have changed the word accordingly (page 21).

13. Change to “prove”: could further **proof** the validity

Reply. The word “proof” has been replaced with “prove” (page 22).

14. Throughout the document, change “z” to “s” (British spelling rather than US spelling). Also, change words like “behavior” to “behaviour”

Reply. The manuscript has been turned to the British spelling.

15. It is advisable to have a professional language edit of the entire document.

Reply. Thank you for the advice. The manuscript has been thoroughly reviewed and edited.

Background:

16. In the Background section, one would expect a more elaborate (and explicit, with the appropriate headings) explanation of the important variables in the dataset, such as neuroticism, general anxiety, test anxiety, math self-concept, language self-concept, math self-efficacy, state anxiety, subject liking and persistence, and math-gender stereotype endorsement. This would help future users of the data to better understand what the variables measured, and how they relate to arithmetic performance.

Reply. We apologize for the inconvenience to the reader. In the paragraph “Individual differences in mathematics anxiety and related constructs”, we explicitly defined the variables that are included in this dataset and how they are related to mathematics anxiety and math performance (page 2). Please note that the limit of the Background is 1000 words (indicated in the paper template), so our explanation had to be concise:

1.3 Individual Differences in Mathematics Anxiety and Related Constructs

Mathematics performance, as well as educational choices, are not solely determined by mathematics anxiety. Other individual factors contribute to this interplay.

1.3.1 Other forms of anxiety

Common (trait) mathematics anxiety exhibits associations with other forms of anxiety (Cipora et al., 2015; Hembree, 1988, 1990; Lunardon et al., 2022; Orbach et al., 2020; Rossi et al., 2023), which are associated with mathematics

performance as well, although their association is weaker compared to that with mathematics anxiety. Importantly, the influences of mathematics anxiety usually prevail, when other forms of anxiety are included in a model (Demedts et al., 2022; Hill et al., 2016). Moreover, mathematics anxiety can coexist with other forms of anxiety in some individuals, while others may experience specific academic anxiety types (i.e., test and mathematics anxiety), with distinct relations to arithmetic performance (Carey et al., 2017; Mammarella et al., 2018; Rossi et al., 2023). The forms of anxiety considered in this context are:

- *state anxiety*: experienced at the very moment of performing a task (Endler & Kocovski, 2001)
- *test anxiety*: experienced in evaluative settings in general (Hembree, 1988);
- *general anxiety*: the general tendency to feel anxious about everyday situations (Hembree, 1990)
- *neuroticism*: the broader tendency to be emotionally unstable (Cipora et al., 2015; Lunardon et al., 2022; Rossi et al., 2023)

1.3.2 Positive attitudes

Individual beliefs and stereotypes regarding personal attitudes and skills, especially in the mathematical domain, can interact with mathematics anxiety. Positive mathematics attitudes negatively correlate with mathematics anxiety (Hembree, 1990; Macmull & Ashkenazi, 2019, Rossi et al., 2022) and positively influence mathematics performance (Marsh et al., 2006; Pajares & Miller, 1994) – in contrast to mathematics anxiety. Relevant attitudes in this context are:

- *academic self-concept*: the degree to which people perceive themselves as proficient in specific academic domains, such as mathematics and language (Goetz et al., 2007)
- *mathematics self-efficacy*: the degree of confidence in one's own capabilities to solve mathematical tasks (Bandura, 2012)
- *subject liking and perseverance*: intrinsic enjoyment experienced when completing a task and the continuous effort despite its difficulty (Pintrich & Schunk, 2002)
- *mathematics-gender stereotype endorsement*: the degree of agreement with the false belief that mathematics is for men and not for women (Blanton et al., 2002).

17. Since arithmetic performance will usually be used as the dependent variable, it may be wise to explain it first. Also placing it in the first column (after the demographic variables) may be a good idea.

Reply. We have explicitly mentioned arithmetic skills at the very beginning of the Background (page 2): “Arithmetic skills, i.e., the ability to solve operations such as additions, subtractions, multiplications and divisions, are fundamental for everyday activities, particularly within educational settings where they are actively taught and learned. People vary in their levels of arithmetic proficiency, and, to understand these interindividual differences, researchers have focused on emotions and attitudes linked to mathematics learning.”

As far as the tables are concerned, we decided not to change the order of the rows and columns because this is meant to represent the order in which the tasks were presented to the participants (see note to Table 3).

Explanation of math load:

18. You state that you have used a 9-point Likert scale. Indicate 1= and 9=. State what the original question was that the respondents had to answer. Do not paraphrase the Likert scale into 3 questions - ("For the choice of my study program the math load played a role because I wanted to avoid math subjects", "For the choice of my study program the math load played a role because I wanted to take math subjects" and "For the choice of my study program the math load played no role").

Reply. We have reported the original question and the labels that participants read on the Likert scale (page 12): "Furthermore, all three samples were asked to indicate how much and in what manner the mathematics load in the study program influenced their study program choice. This was assessed by a single item ("For the choice of my study program the mathematics load played ...") on a 9-point Likert scale (1 = "... a role because I wanted to avoid mathematical subjects", 5 = "... no role", 9 = "... a role because I wanted to take mathematical subjects"). Lower values indicate that the mathematics load mattered in the sense that led them to avoid mathematics courses. Medium values indicate that mathematics load did not play any role in the choice of the study program. Higher values indicate that the study program was chosen because of the willingness to pursue mathematics courses."

Mental arithmetic ensured?

19. You state that "Participants were instructed to solve the problems in the presented order, mentally and without using a calculator." However, the test was done online. How can you be sure that participants truly used only their mental capacity?

Reply. The instruction stated explicitly that the task should be conducted without a calculator. Indeed, in an online study, compliance with instructions cannot be controlled for. In the end of the survey, questions were employed to ensure data quality (e.g., dishonest completion of the survey leads to data exclusion), and correlations between mathematics anxiety and arithmetic performance were found, indicating that compliance with instructions can be probably assumed for most of the participants. However, the online study is limited in this respect and therefore we state the limitation of the study (page 23): "Online data collection is limited in the point that compliance with instructions cannot be assured (e.g., mental arithmetic without any aids). At the same time, participants were not incentivised for high performance."

Ethics:

20. In 2.4, you mention that participants were offered to enter a raffle for vouchers. Yet, in 2.7, you state that data collection was anonymous. How could participants then receive their vouchers?

Reply. The email addresses for the raffle were stored separately from the collected data and therefore could not be linked. Entering the raffle was not mandatory, and participants, who wanted to remain fully anonymous could simply do not provide a contact email address. We have added this piece of information as a footnote in the text (page 5).

Exclusion criteria:

10.1 Exclusion 1

21. You state that: “The score for arithmetic performance was operationalized as the number of correctly solved problems. The final score as well as the responses and accuracy per item **were retained only for participants who did not skip any item** (see Quality control).” If a participant skipped an item, it could be that he/she struggled with the question and chose to skip it. Removing these observations may lead to valuable information being lost. Would it not be better to include these observations, but add another variable “Completed all questions” which could be either true or false? Then a researcher can choose to use them or omit them.

Reply. We agree that this data might be useful for some specific research questions and researchers should decide how to deal with these special cases. Therefore, we now provide the data of all participants regardless of whether or not they skipped an item and excluded the skippers only from the sum score for arithmetic performance as we believe the score is problematic. However, if the users of the dataset wish to use it nevertheless, it is easy to calculate it based on responses to single items. We changed the description in the main text accordingly (page 14): “The score for arithmetic performance was operationalized as the number of correctly solved problems. The final score was retained only for participants who did not skip any item (see Quality control)”.

10.2 Exclusion 2

22. “We excluded from the dataset participants who declared not to have responded honestly, who reported a very noisy or extremely noisy environment, and **who took more than 30 minutes to complete the survey**”. Why were these observations excluded? Would it not be better if they were included but the time it took to complete it was included in the dataset?

Reply. On average, completion of the survey lasted about 15 min. Participants taking more than twice as long very likely got distracted during study participation. In some cases the recorded time was several hours indicating that the participant either left the computer or moved to other activities. In such cases we cannot assume sufficient data quality. Therefore, these participants should be excluded from the dataset. This only affected $n = 24$ participants across all samples. In the manuscript, we added this explanation in brackets: “i.e., more than $2 \times$ typical study duration” (page 16).

Introducing Table 3

23. In the first paragraph of 2.5, you refer to Table 3. Yet Table 3 only appears a few pages later. The reader gets lost when a table just suddenly “appears, without a linking sentence.

Reply. We have moved Table 3 earlier, just after the first paragraph of section 2.5.

Table 4

24. Some of the correlations in Table 4 are highlighted with an asterisk (*) I assume they were significant at 5%? Please indicate. Consider using ** for significance at 1%.

Reply. We apologise for the inattention. We have specified the significance levels in the caption (page 17): “Significance levels are specified as follows: * $p < .05$; ** $p < .01$; *** $p < .001$.”

25. The table is confusing. Usually, the top row contains the correlations and the bottom row contains the probabilities. Here, some cells contain the upper value and others the lower value. Please explain what the upper values and lower values are.

Reply. All values in the table are correlations. Each correlation appears only once. As outlined in reply to comment 24, we used asterisks for highlighting different significance levels. Moreover, some values are very small, this might have been misleading. These points and the significance levels are now explained in the caption of the revised table (page 17): “Correlations between the study variables. Significant correlations are marked with asterisks directly under the correlations. Significance levels are specified as follows: * $p < .05$; ** $p < .01$; *** $p < .001$. Note that some correlations are very small.”

Reviewer B

I choose to upload my review in form of article with tracked changes. Hence, for details, check the uploaded reviewed articles. Nonetheless, although I find both the article and its accompanied dataset interesting and worth publishing, here are just a few key points to consider to help improve manuscript layout;

(1) Section 1 sub-titles could be numbered as well, beginning with 1.1, 1.2, etc.

(2) Under methodology, particularly on sampling and sample sizes, it is not clear how the initial samples sizes were arrived at. Were there any statistical sampling techniques employed to arrive at those initial sample sizes before using the inclusion-exclusion criteria?

(3) Again, under methodology, data description, there seems to be some little confusion between study populations described in the manuscript and as captured in the manuscript title. Whilst the manuscript title simply states, “.....university teachers and students..”, the manuscripts further demarcates these as “university students” and “elementary school teachers”. I am not sure if the context is the same. Just try to reconcile the two or provide clarification.

(4) On survey instruments, whilst I found the answer to my earlier question on availability or not of German-English version instruments in the study limitation paragraph, I should still think we have the original German and German-French tools. Is there a chance to include these within the data repository to promote the replica of research methodology?

(5) On study location, I guess a mention of demographic structure, largely urban, could be of help for those who have less knowledge of study population's location. Perhaps, inclusion of a map would also be of help.

Reply. Thank you for taking time for reviewing the manuscript, your positive evaluation and constructive feedback. We have copied reviewer B's comments to the manuscript here and replied to each of them. The key points stated here have been addressed in the specific comments below. Changes made directly to the text have been included in the revised version of the manuscript.

Abstract

1. “*The present dataset includes math anxiety*”. “...include measures of math anxiety...”

Reply. We have modified this sentence in the abstract accordingly.

Keywords

2. “*teachers*”. “teachers and university students”

Reply. We have added the keywords “university students” and “primary school teachers”.

Background

3. I suggest all subsections of section (1) of the manuscript should be numbered, beginning with 1.1 “On the importance of Math Anxiety”

Reply. Thank you for this suggestion. We have numbered all subsections of the “Background” section.

4. “*Do primary school teachers exhibit different mechanisms compared to students in other study programs?*”. Relevant research question. However, could it be paraphrased to suit the data set in context? I mean we are posing research questions, perhaps answerable by the data set in mention which came from University teachers and students. Hence, I thought we could stick to such domain of teachers for now.

Reply. We are sorry for the confusion. The teachers in the present dataset are indeed primary school teachers, either in-service (working as primary school teacher) or pre-service (studying primary school education at University). The latter are enrolled at the university and are studying to become primary school teachers, so in the manuscript we have conceptually included them in the group of “teachers”, not in the group “university students”. However, this does not prevent future users of the dataset to include pre-service teachers in the group of university students, if their research question makes this more convenient. In fact, in the dataset there are two variables that specify at what stage the teachers are: the variable “teacher_stage” (levels: “primary school teacher”, “study”) and the variable “teacher_experience” (levels: “in-service”, “pre-service”).

Since age of entrance and duration of primary school vary across countries, we have specified this information in the main text (page 5): “The teachers enrolled in the study were working or preparing to work with children who enter schooling in their respective countries, which happens at the age of six. Primary school consists of grades 1-4 in Germany and grades 1-6 in Belgium.”

Considering your next comments, we acknowledge that the title was not precise. Therefore, we have added the specification “*primary school teachers*” also into the title: “The AMATUS dataset: Arithmetic performance, Mathematics anxiety and Attitudes in primary school Teachers and University Students.”

5. “*The sample consists of university students and elementary school teachers*”. Which is which on category of teachers? The study title says “University teachers and students”, here we have university students and elementary school teachers...

Reply. The first version of the title said “The AMATUS dataset: Arithmetic, Math Anxiety, Math Attitudes and related constructs *in Teachers and University Students* with different mathematical background”. However, we acknowledge that it was not precise and could lead to misunderstandings. Therefore, we have worked on the title (see reply to comment 4).

6. *“allows comparing teachers and non-teachers,”* I don’t know how best we can include the “non-teachers” concept in the title to clearly reflect the content of the data set. I do not have immediate suggestions yet, but this can be guided by journal editors

Reply. We hope that the new version of the title (see response to comment 4) is clearer.

Time of data collection

7. *“Data for Sample 1 was collected in June 2017, for Sample 2 in January 2018, and for Sample 3 from May 2018 until December 2018.”* I guess this its proper to quantify it as panel datasets to prepare the user for a few proper research design when investing research questions. Considering also the differences in human subject backgrounds.

Reply. Thank you for this suggestion. However, the present study cannot be classified as a panel study (as we understand it <https://doi.org/10.1016/B0-08-043076-7/00748-8>). Firstly, the design is not longitudinal; the three samples consist of different individuals. Moreover, the groups are not uniform and include individuals with specific characteristics, such as being teachers or students. Additionally, as we mention below, our samples were convenience samples, so their structure is not representative to the structure of the population.

Location of data collection

8. Since the dataset is sharable publicly and reusable by several researchers at global level, it would be plausible to indicate that respondents were drawn from urban areas, assuming that is correct since the variable code book does not have a socio-demographic variable of residence (urban/rural). Moreover, with an inclusion of a map to indicate actual geographical locations.

Reply. This is an interesting suggestion. Unfortunately, we do not have exact information about participants’ residence. The institutions that recruited the participants have students from across Germany and Belgium. We can infer that at the time of data collection they lived in the proximity of the institutions where they studied or worked – but students and the in-service teacher came from all over the country. Moreover, we cannot exclude that participant came from different geographical backgrounds. We have added this information in the text (page 4): “No information about the participants’ residence (urban/rural) is available.”

Sampling, sample and data collection

9. The section lacks clarity on how the sample sizes were arrived at. Could we describe how we arrived at these samples 1, 2, and 3? Any sampling technique or methodology in play?

Reply. We tried to reach as many participants as possible from the participating institutions and by other contacts. We did not predetermine sample size. We have specified the sampling technique we used at the beginning of the section “Sampling, sample and data collection” (page 4): “For each sample, we employed convenience sampling and then excluded participants that did not satisfy the eligibility criteria described below. Table 1 shows the total number of participants involved in the study and the number of participants excluded from the dataset.”

10. "The initial sample consisted of 1404 participants (1049 for Sample 1, 164 for Sample 2, and 191 for Sample 3)." How did we arrive at this initial sample sizes, before employing the inclusion-exclusion criteria in the follow up process to get to final sizes?

Reply. see comment 9

Table 1

11. For consistency sake with descriptors used in the dataset or codebook, use "N/A" for no applicable response

Reply. We have replaced "n.a." with "NA", using the same notation as in the dataset and the codebook, in Tables 1 and 3. We used "NA" instead of "N/A" because it is a default in R for empty data cells.

Materials/Survey instruments

12. "originally formulated in German and then translated into French by a German-French bilingual". Is there, by any chance, to have access to the English version of the survey instrument? Just being curious for a wide range of re-usability purposes. I guess these survey instruments are also going to be submitted alongside the revised manuscript

Reply. The codebook includes the French and German items that we used in the survey, as well as their English version. The English version consists of the original items, when the German and French items were translated from English or an equivalent version of the same questionnaire exists in English, or in our translation in the case of the items created ad-hoc for the study.

We specified that the English version of the items can be found in the codebook (page 21): "The codebook indicates the constructs assessed, the instruments used, the variable names in the dataset, a description in English of the variables, the type of variables, whether the item score was reversed, *the English version of the items*, the used German and French versions of the items, and in which sample each variable was assessed."

Quality control

13. "for each measure." Just confirm. I guess not all measures were subjected to these tests, particularly those you have indicated n.a (not applicable).

Reply. You are right. We did not compute reliability for one-item measures. We have added this information also at this point of the text (page 16): "Moreover, we computed reliability for each measure which consists of more than one item [...]."

Table 4

14. I guess you needed to add a footnote to this Table 3, on interpretation of those measures with stars, "significantly correlated at alpha=5%", if that's the level of confidence used.

Reply. We apologise for the inattention. We have specified the significance levels in the caption (page 17): "Significance levels are specified as follows: * $p < .05$; ** $p < .01$; *** $p < .001$."

Data anonymisation and ethical issues

15. *“Finally, given the high variability in age in the teacher samples, we added an age range variable in the dataset and removed teachers’ age in years.”* Was this done in course of survey or before the survey was rolled out? Just wondering how you may have handled the two different variables at the data cleaning stage for those participants who may have earlier replied to “single value age” and others who accessed the same questionnaires but replying to “age ranges”.

Reply. The change from age in years to age range was done after the end of data collection, at the step of data preprocessing (see also the file AMATUS_preprocessing.pdf on OSF, <https://osf.io/gszpb/>). We have stated this more explicitly a few lines before, as all changes listed in this paragraph were done during data preprocessing (page 20): “The risk of identification of study participants was reduced by removing free-entry responses and categorization in *the data preprocessing step*”

Existing use of data

16. *“In addition, there is a paper in preparation, which was preregistered for secondary data analysis”.* It may also be good to mention, among the provided possible research question, which this second paper is addressing to, avoiding any potential research duplication from any user who accesses this dataset later.

Reply. Thank you for this suggestion. We have added a brief description of this study (page 20): “In addition, there is a paper in preparation, which was preregistered for secondary data analysis (<https://osf.io/z5egb/>). This study will employ Latent Profile Analysis (LPA) to investigate whether different combinations of anxiety types and math attitudes can differentiate university students in programs with different mathematics load.” Interested readers might find all additional details in the shared preregistration form.

Object/file name

17. I will ask again, do we have an opportunity to get the survey instruments deposited into the repository as well?

Reply. Of course. As stated in response to comment 12, the codebook in the OSF repository (<https://osf.io/gszpb/>) contains the items in French, German, and English.

Reuse potential

18. *“What is more, for more.”* Consider language revision

Reply. We have replaced “What is more” with “Additionally” (page 22) and we have checked the whole document for language inaccuracies.

19. *“Although translation was carried out by native speakers, the two versions may not be perfectly equivalent. Finally, the data was collected in Germany and partially in Belgium so that it might be limited to these educational systems, cultural backgrounds, and languages. To generalize the findings based on the AMATUS dataset, we invite researchers to translate and use the measures of this study all over the world in order to compare the resulting findings and check replicability.”* Well, I guess my earlier posed questions on English version instruments have been answered by this paragraph. Thanks.

Reply. We are grateful for your evaluation. We refer you to the codebook stored in OSF (<https://osf.io/gszpb/>) for the English version of the questions.